Estimation of the probability of daily fluctuations of incidence of COVID-19 according to official data

http://orcid.org/0000-0003-4549-7172 Gerasimov Andrey 1 andr-gerasim@yandex.ru
Galkina Elena 1
Danilova Elena 1
Ikonnikova Irina 1
Novoselova Tamara 1
http://orcid.org/0000-0003-0587-1609 Orlov Yuriy L. 2
Senenycheva Irina 1
1 Department of Medical Informatics and Statistics, I.M. Sechenov First Moscow State Medical University , Moscow , Russia
2 Institute of Digital Medicine, I.M. Sechenov First Moscow State Medical University , Moscow , Russia
Hochheiser Harry
Electronic publication date: 2021 Jun 4
Publication date: 2021
Volume: 9
Electronic Location ID: e11049
Received 2020 Sep 14; Accepted 2021 Feb 10
Copyright: © 2021 Gerasimov et al.
Copyright year: 2021
Copyright holder: Gerasimov et al.
License: This is an open access article distributed under the terms of the Creative Commons Attribution License, which permits unrestricted use, distribution, reproduction and adaptation in any medium and for any purpose provided that it is properly attributed. For attribution, the original author(s), title, publication source (PeerJ) and either DOI or URL of the article must be cited.
License URL: https://creativecommons.org/licenses/by/4.0/

Keywords: COVID-19, Random fluctuations of incidence, Mathematical methods of morbidity analysis, Analysis of daily morbidity, Analysis of morbidity dynamics, Random fluctuations in mortality, Expected distribution of incidence

Funding: The authors received no funding for this work.

==============================
When studying the dynamics of morbidity and mortality, one should not limit ourselves to analyzing general trends. Interesting information can be obtained from the analysis of deviations in morbidity and mortality from the general dynamics. Comparison of the cases of morbidity or death for adjacent time intervals allows us to find out whether the changes in conditions were for short periods of time and whether the cases of morbidity or death were independent. The article consists of two parts: Study of the probability distribution (CDF) of the difference between two independent observations of the Poisson distribution; Application of the results to analyze the morbidity and mortality trends by day for the new coronavirus infection. For the distribution function of the module of difference between two independent observations of the Poisson distribution, an analytical expression has been obtained that allows to get an exact solution. A program has been created, whose software can be downloaded at http://1mgmu.com/nau/DeltaPoisson/DeltaPoisson.zip. An approximate solution that does not require complex calculations has also been obtained, which can be used for an average of more than 20. If real difference is greater than expected, it may be in the following cases: morbidity or mortality varies considerably during the day. That could happen, for example, if the registered number of morbidity on Saturday and Sunday is less than on weekdays due to the management model of the health system, or if the cases are not independent; for example, due to the active identification of infected people among those who have come into contact with the patient. If the difference is less than expected, it may be due to external limiting factors, such as a shortage of test systems for making a diagnosis, a limited number of pathologists to determine the cause of death, and so on. In the analysis of the actual data for COVID-19 it was found that for Poland and Russia, excluding Moscow, the difference in the number of cases and deaths is greater than expected, while for Moscow—less than expected. This may be due to the information policy—the effort to somehow reassure Moscow’s population, which in the spring of 2020 had a high incidence rate of the new coronavirus infection.

Introduction

The epidemic process of infectious diseases is a random process (Black et al., 2009; Krause et al., 2018; Simões, Telo da Gama & Nunes, 2008).

Even for very large populations, numbering in the millions or more, the average number of infected people per day is small, and even under stable conditions, random fluctuations in the number of infected people gradually accumulate. Its increase is limited to changes in the level of collective immune status, but this process is slow. The magnitude and nature of random fluctuations in the proportion of infected people can be studied using stochastic options of compartment models of the epidemic process, such as SIS, SIR, SIRS, SEIR, and so on (Liu et al., 2019; Nakamura & Martinez, 2019). Within these models, it turns out that stochastic effects generate morbidity waves with a period from several years to several decades, depending on the duration of the disease, its infectivity and so on.

However, along with these, there is also a fluctuation in morbidity and mortality, which manifests itself in short periods, including at the level of days. They may be related not only to drastic changes in the epidemic process, but also to changes in the registration conditions (for example, fewer cases are detected on weekends), as well as to the fact that the detection of cases is not necessarily unrelated events. For example, several cases may be detected during a school survey.

There may also be multiple cases detected simultaneously in the family.

Therefore, the analysis of changes in morbidity and mortality over short time periods can reveal the features of the organization of morbidity detection.

An integral part of the analysis of actual morbidity data is to assess the correctness and comparability of official reporting data (Cooper, Bynum & Somers, 2009; Isanaka et al., 2016)

Differences in incidence may be related not only and not so much to differences in disease risk, but to differences in case detection and reсording. Therefore, if in the first region the incidence is higher than in the second under similar conditions, it does not mean that doctors perform worse in the first region, the situation is often the opposite one.

Detecting a case and making a diagnosis is not something unambiguous.

First, for almost all infectious diseases, the proportion of manifest cases is low. Most cases are asymptomatic or unclear symptomatic, and for the most part remain undetected. Some cases are not detected by referrability, but by active detection efforts in foci and among at-risk groups (Abbott et al., 2017; Leung, Trapman & Britton, 2018).

Second, incidence depends on diagnosis criteria. For example, an infectious disease can be diagnosed both on symptoms and in case of presence of laboratory confirmation. In the latter case, a large part of them goes recorded as “acute respiratory disease” and “intestinal infection of unclarified epidemiology”.

The criteria for diagnosis can vary over time and between countries. For example, the decrease in the incidence of tuberculosis in Russia at a rate of about 10% observed in the last decade is a consequence of the constant change in diagnosis criteria with their fitting to WHO criteria, where only cases with active bacteriodisposition are considered as tuberculosis, while maintaining constant criteria for diagnosing, the incidence of tuberculosis in Russia would continue to increase (Yu et al., 2019; Gerasimov & Mikheeva, 2018).

Fluctuations in morbidity can also be associated with organisational aspects. For example, in Brazil, the number of COVID-19 cases detected on Saturday and Sunday is about one and a half times lower than on weekdays. In Russia, there are no differences in the incidence of COVID-19 on weekdays and Sundays, but there is a three times difference in the number of recovered due to the fact that on Saturday and Sunday there are no dismisses from hospitals.

Unfortunately, among the factors influencing official morbidity, there is also a desire to show a picture better than it is in reality; to hide flaws and errors and “report nicely”. This is usually found in a decrease in morbidity and mortality. An example is the legendary statement of the Belarusian leader that “we have no deaths from COVID, we have deaths with COVID”.

However, an analysis of official COVID-19 incidence data revealed another phenomenon: an overly stable incidence, in which the number of cases detected per day hardly changes.

Thus, the traditional epidemiological analysis of the morbidity and mortality rates, designed to seek trends, should be supplemented with an analysis of the severity of differences in the number of cases and deaths over the following intervals. This can provide information about the special characteristics of the epidemic process, as well as about the performance of the identification and registration system of cases and death.

Materials and Methods

Let A be the number of cases detected over a period of time, including per day. Then if the cases are independent and the number of cases is low compared to the overall population, then A is distributed by Poisson (Gerasimov, 2014). For the Poisson distribution, variance equals mathematical expectation. Therefore, if x1, x2 are two independent observations of the Poisson distribution with the same average λ, then E(x1−x2)=E(x1)−E(x2)=λ−λ=0 and D(x1−x2)=D(x1)+D(x2)=λ+λ=2λ.

Besides, for a sufficiently large mathematical expectation, the Poisson distribution is close to the normal distribution (Stuart & Ord, 2009). Therefore, when increasing the mathematical expectation of the Poisson distribution, the distribution of value (x1−x2) tends to a normal distribution with a mathematical expectation equal to zero and variance, equal to 2λ. Consequently, when increasing λ, the distribution approximates distribution 2λχ12,

However, the mathematical expectation of the number of sick is unknown to us. If x1, x2 are two independent observations of the Poisson distribution with mathematical expectation λ, then Δ=(x1−x2)2x1+x2 is not distributed as χ12, since, first, the Poisson distribution is not exactly the same as normal and secondly, in expression (x1−x2)2x1+x2 the same values x1, x2 are used both to estimate variance, and to estimate mathematical expectation.

The value of distribution function Δ=(x1−x2)2x1+x2 can be calculated as

(1) FΔ(x)=∑k≥0,n≥0,k+n>0,Δ≤xe−kk!e−λe−nn!e−λ=∑k≥0,n≥0,k+n>0,Δ≤xe−k−nk!n!e−2λ

where Δ=(k−n)2k+n, k, n are the natural numbers, FΔ()—distribution function (cumulative density function) of Δ.

Since it requires a fairly large amount of calculations, a program has been created, whose software can be downloaded at http://1mgmu.com/nau/DeltaPoisson/DeltaPoisson.zip.

Figure 1 below shows the calculated distribution function of Δ and the value χ12 distribution function. It can be seen that the calculated distribution functions differ very little from the “chi-squared”—distribution even for a small λ.

Figure 1 Distribution functions for value Δ for λ = 3, 10, 30 and for χ2 distribution.

Further increase of λ does not change the shape of the distribution, it only becomes smoother, close to continuous, the magnitude of the spikes decreases.

It results into a conclusion that to assess the probability of differences in incidence over time intervals, a sufficiently accurate estimate of the average incidence is not required, since the value λ for a not very small absolute incidence has little effect on the distribution under study.

Results

Data on the number of people who became ill and died from COVID-19 were taken from publicly available sources: https://covid.observer/ (Data aggregated by the Johns Hopkins CSSE) and https://xn--80aesfpebagmfblc0a.xn--p1ai/ (Official data for the Russian Federation).

We used data for Russia and Poland for the period from July 1 to August 12, since during this period there was no rapid change in incidence (Fig. 2).

Figure 2 Number of cases and deaths from COVID-19 for Russia (separately: Moscow and the all other regions) for the period from 1 July to 12 August and Poland for the period from 2 April to 27 May.

Figure 2 Number of cases and deaths from COVID-19 for Russia (separately: Moscow and the all other regions) for the period from 1 July to 12 August and Poland for the period from 2 April to 27 May.

Analysis of the severity of differences in morbidity and mortality for Moscow are given in Table 1. Table 1 shows that close values are too common for morbidity data for adjacent days. In particular, the values Δ that should have been present with the probability FΔ < 0.1 were observed in 11 cases out of 42, while the probability that the binomial distribution with N = 42 and P = 0.1 takes values of 11 or more is only 0.23%.

Table 1 The number of COVID-19 cases and deaths by day in Moscow from July 15 to August 12, the magnitude of the differences in incidence ∆ for neighboring days and the probability FΔ that such or a lower value may be accidental.

Date	Number of new cases per day	Δ	FΔ(), the exact solution according to formula (1)	FΔ (), approximate solution according to χ21 distribution	
Ill	Dead	Ill	Dead	Ill	Dead	Ill	Dead	
16.07.2020	531	24	8.118	0.472	0.9956	0.5106	0.9956	0.5078	
17.07.2020	575	13	1.750	3.270	0.8143	0.9303	0.8142	0.9295	
18.07.2020	578	14	0.008	0.037	0.0705	0.1569	0.0704	0.1526	
19.07.2020	591	14	0.145	0.000	0.2963	0.0757	0.2962	0.0000	
20.07.2020	578	15	0.145	0.034	0.2963	0.1513	0.2962	0.1473	
21.07.2020	602	17	0.488	0.125	0.5154	0.2803	0.5152	0.2763	
22.07.2020	638	19	1.045	0.111	0.6935	0.2646	0.6934	0.2611	
23.07.2020	608	14	0.722	0.758	0.6047	0.6200	0.6046	0.6159	
24.07.2020	645	11	1.093	0.360	0.7042	0.4573	0.7041	0.4515	
25.07.2020	648	14	0.007	0.360	0.0666	0.4573	0.0665	0.4515	
26.07.2020	683	9	0.920	1.087	0.6627	0.7070	0.6626	0.7029	
27.07.2020	694	13	0.088	0.727	0.2332	0.6127	0.2331	0.6062	
28.07.2020	674	10	0.292	0.391	0.4114	0.4744	0.4113	0.4684	
29.07.2020	671	13	0.007	0.391	0.0653	0.4744	0.0652	0.4684	
30.07.2020	678	12	0.036	0.040	0.1512	0.1631	0.1512	0.1585	
31.07.2020	695	14	0.210	0.154	0.3537	0.3102	0.3536	0.3051	
01.08.2020	690	13	0.018	0.037	0.1070	0.1569	0.1069	0.1526	
02.08.2020	664	12	0.499	0.040	0.5203	0.1631	0.5202	0.1585	
03.08.2020	693	13	0.620	0.040	0.5690	0.1631	0.5689	0.1585	
04.08.2020	691	12	0.003	0.040	0.0430	0.1631	0.0429	0.1585	
05.08.2020	687	11	0.012	0.043	0.0859	0.1702	0.0858	0.1652	
06.08.2020	684	13	0.007	0.167	0.0647	0.3224	0.0646	0.3169	
07.08.2020	686	12	0.003	0.040	0.0432	0.1631	0.0431	0.1585	
08.08.2020	691	14	0.018	0.154	0.1073	0.1631	0.1072	0.3051	
09.08.2020	689	12	0.003	0.154	0.0430	0.1631	0.0429	0.3051	
10.08.2020	694	13	0.018	0.040	0.1070	0.1631	0.1070	0.1585	
11.08.2020	694	14	0.000	0.037	0.0107	0.1569	0.0000	0.1526	
12.08.2020	689	12	0.018	0.154	0.1070	0.1631	0.1070	0.3051	

There are also valid differences in the Δ distribution as well.

When analyzing data on Russia (exclusing Moscow) and Poland, we see the following

The distributions of observed values, as has already been noted, differ from the expected uniform distribution, but the nature of differences is not the same:

From Fig. 3 it follows that for Moscow, both for the number of cases and for the number of deaths, small values of Δ are more often than expected, while for Russia excluding Moscow and Poland, large values of Δ are more often than expected.

Figure 3 Empirical distributions of value F∆ for Moscow, Russia (excluding Moscow) and Poland.

It follows from Table 2 that: For the considered period of relatively stable morbidity, the median of the number of cases of COVID-19 per day in Moscow was 671 cases, the median of incidence differences Δ is 0.163, corresponding to FΔ = 0.313. For the number of deaths from COVID-19 per day we have a median of 14 cases, the median of differences for neighboring days Δ is 0.154, which corresponds to FΔ = 0.305. That means, for both the number of cases and the number of deaths, close values for neighboring days were more frequent than expected,

If one conducts a similar analysis for Russia except Moscow, the median of the number of cases was 5,240, the median of the differences for neighboring days Δ was 0.587, which corresponds to FΔ = 0.555. For the number of death, the median is 127, the median Δ is 2.777, corresponding to FΔ = 0.904.

Table 2 Characteristics of the number of COVID-19 cases and deaths over the periods under review.

	Moscow	Russia excluding Moscow	Poland	
Ill	Dead	Ill	Dead	Ill	Dead	
Increase rate per day, %	0.26%	−2.91%	−0.89%	−0.97%	−0.06%	−0.88%	
p (comparison of FΔ distribution with uniform, Kolmogorov–Smirnov criterion)	0.006	<0.001	0.127	<0.001	<0.001	<0.001	
Median	
Number of cases per day	671	14	5240	127	337	13	
Δ	0.163	0.154	0.587	2.777	5.075	1.000	
FΔ	0.313	0.250	0.555	0.904	0.976	0.701	

If accepted assumptions correspond to the truth, the value FΔ, like any distribution function, must be evenly distributed. When comparing the obtained distributions with uniform using the Kolmogorov–Smirnov criterion, it can be seen (Table 2) that for morbidity in Moscow the difference is true with p = 0.006, for Russia excluding Moscow differences are unreliable, p = 0.126. At the same time, for mortality in both Moscow and Russia without Moscow, the differences in the actual distribution from the expected are true with p < 0.001.

At the same time, for Poland, the differences in both the number of cases and the number of deaths in neighboring days are higher than expected with p < 0.001.

Discussion

When analyzing the infectious morbidity, one of the characteristics is a focality, that is, the degree of grouping of individual cases, which can be a consequence of family hotbeds of disease, foci in organized children’s groups, etc. So, if cases are detected not independently, but by N cases at once, it increases the incidence by N times, and the variance by N2 times, that is, the ratio between the variance of the number cases and the number of cases can give an estimate about the size a foci.

The assumptions about the independence of individual cases are not entirely accurate, as both the causes of disease and their identification extend the effect not on one person, but on a group of individuals. This is especially pronounced for infectious diseases, as the emergence of a source of the pathogen increases the risk of disease for many contacts, and the detection of one case leads to more active identification among those in contact with him/her. Identifying one case increases the likelihood of detecting other cases, so the variance in the number of cases of disease should be greater than the mathematical expectation.

Also, the variance of morbidity might be increased by changes in the conditions of the epidemic process over time, in which the mathematical expectation of the number of cases and deaths varies over days.

Conclusion

The change in the conditions of the epidemic process, detection of cases, the criteria for making a pristine and posthumous diagnosis, and the fact that for infectious diseases individual cases are not independent increase the incidence differences over adjacent time intervals. Therefore, in the analysis of the actual data, fluctuations of incidence are expected, greater than according to the distribution (1).

In the analysis of the actual incidence of COVID-19 in Poland, an increase in the difference in the number of cases and deaths in neighboring days was found with p < 0.001. However, for data on the number of cases and deaths in Moscow, on the contrary, the difference in the number of cases and deaths in neighboring days is less than expected showing p < 0.001, whereas for the data for Russia, with the exception of Moscow, no reliable differences from those obtained under the assumption of constancy and independence of cases of the disease were revealed.

It follows that at least for Moscow there is a deliberate smoothing of actual morbidity and mortality data, perhaps to reassure the population.

A few days after finishing work on the preliminary text of the article, one of the authors briefly mentioned on his blog that there are signs of manipulation of the data on morbidity and mortality from COVID-19—the difference in the number of cases for adjacent days is too small. Three days after that, starting from August 23, the differences in the number of cases of COVID-19 cases in Moscow over the next few days increased many times and began to correspond to the expected.

Supplemental Information

Supplemental Information 1 Raw data on number of cases and deaths by the regions.

Raw data on number of COVID-19 cases and deaths by the regions in March-August 2020.

Click here for additional data file.

Additional Information and Declarations

Competing Interests

Author Contributions

Data Availability

Yuriy Orlov is an Academic Editor for PeerJ.

Andrey Gerasimov analyzed the data, prepared figures and/or tables, authored or reviewed drafts of the paper, and approved the final draft.

Elena Galkina analyzed the data, authored or reviewed drafts of the paper, and approved the final draft.

Elena Danilova analyzed the data, authored or reviewed drafts of the paper, and approved the final draft.

Irina Ikonnikova analyzed the data, authored or reviewed drafts of the paper, and approved the final draft.

Tamara Novoselova analyzed the data, authored or reviewed drafts of the paper, and approved the final draft.

Yuriy L. Orlov analyzed the data, authored or reviewed drafts of the paper, and approved the final draft.

Irina Senenycheva analyzed the data, authored or reviewed drafts of the paper, and approved the final draft.

The following information was supplied regarding data availability:

Link to used morbidity and mortality data is available at Yandex (no registration required): https://yandex.ru/covid19/stat. As of January 6, 2021, all used data remains available and has not been corrected.

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
