# Peer review of "Estimation of the probability of daily fluctuations of incidence of COVID-19 according to official data"

_PeerJ, doi:10.7717/peerj.11049_

## Round 0.1 · original submission · Minor Revisions

Both reviewers provided textual critiques of this interesting paper. Most of the comments should be easily addressed with some minor editing and clarification, particularly given the statistical methods and tools. Please pay careful attention to Reviewer 1's comments about the need for clearer framing and exposition regarding the knowledge gap and the motivation for the paper.

·

Basic reporting

The abstract :
The abstract doesn’t describe the idea and the objective of the study.
There is no clear summary about the procedure, steps and outcome of the study.
It needs more details.

Introduction : Needs more details.

Lines: 102-108 needs sentences reconstructions to provide more descriptions

Figures
- Should be named appropriately: example (Figure 1 : )
- All figures should be mentioned inside the text

Tables
- Should be named appropriately: example (Tale 1 : )
- All tables should be mentioned inside the text

The English language
The language used is professional throughout the text except the points that was mentioned above and the highlighted lines in (Red color).
The grammar is observed to be checked throughout the text.
The construction of sentences is good but needs a final revision for more editing.
The overall rating is acceptable to some extent.

Experimental design

Materials and methods : In lines 71-72, the expectations of the two independent observations x_1,x_2 should be equal to get the zero result, I suggest that the author should add this assumption.

Validity of the findings

The study possess a novelty but the author should follow appropriate framing (the problem statement was not clear) to suit the criteria of the journal.

Additional comments

I suggest that the author should :
- improve the description to provide more justification for the study.
- improve the description mode at lines 30- 65, by expanding upon the knowledge gap being filled to give more justification for the study

·

Basic reporting

The article represents important progress for the current situation. The work of the authors is very well focused on computing statistics and it could represent a professional framework that can be applied as a "standard" for other directions (related to COVID-19) and researches.
The abstract doesn’t point out the (general) objective and scope of the article. Suggestion: "In this article, we will…, The article will (examine, present, introduce, etc.)". Need to be improved.
- Line 115: “in table No.1”…Capital “T”
- Line 137: Capital “F” for “…fig. 3…”
- Line 141: Capital “T” for “…table 2…”
- References list should be improved. Nothing from 2020?!

Experimental design

Which tool has been used for computing the distributions (Statistica, IBM SPSS, MatLab, Microsoft Excel?!?!)?
How the analysis of the severity of differences in morbidity and mortality has been computed? The question is: how did you find the probability F_Δ<0.1?

Validity of the findings

Discussion based on the results should be improved as well as mentioning the methods used for computing the distributions. The article will have a more trusted validity for the community once the tools are listed and mentioned in the article.

Additional comments

The article subject is very important and deserves special attention to some of the details, especially about the computations tools used for computing different statistics.
It is very good progress and I strongly encourage you to continue your research by creating and introducing more statistics-oriented software tool analysis. Using the proper tools with respect for statistics, such as STATISTICA, IBM SPSS, MatLab or Excel, will increase the scientific validity and suitability.

---

## Round 0.2 · Minor Revisions

Thanks for your revision. Although the second reviewer is satisfied with this improved version, the first reviewer raises some question about the clarity of the abstract and the presentation of the results. I agree with these comments. Specifically, I suggest a few changes:

1. The abstract should be revised as per Reviewer 1's comments.

2. The introduction should be augmented with a final paragraph describing the goals and outcome of the proposed analyses.

3. The methods should be revised to include a description of how the proposed methods answer the question introduced in the introduction.

4. More detail about the analysis code is needed: were established packages used? Is the source code available?

Thanks for these revisions.

·

Basic reporting

The article contains a novelty in its goal and has a clear problem that needs mathematical techniques and tools to be revealed . The abstract needs to refitted to include the goals and the author work, and should point to the study outcomes. The results should be more expanded to reflect clear fitting of the graphs to the presented data , table and graphs. For the statistical analysis , the author should use advanced package to give a deep results of reflection.

Experimental design

The research bounds within Aims and Scope of the journal. The research question well defined, relevant & meaningful. It is stated how research fills an identified knowledge gap, but it needs more revision to suit the mathematical objectives.

Methods is not described with sufficient detail & information to replicate.it needs to reestablished

Validity of the findings

All underlying data have been provided; they are robust, statistically sound, but data analysis needs more advanced statistical package with more implementation to the reality.

Additional comments

please kindly do the mentioned editting

·

Basic reporting

All the observations were fixed and remediate accordingly.

Experimental design

All the observations were fixed and remediate accordingly.

Validity of the findings

All the observations were fixed and remediate accordingly.

Additional comments

All the observations were fixed and remediate accordingly.

---

## Round 0.3 · Minor Revisions

Thank you for your revisions. This version is much improved and addresses most reviewer concerns adequately. However, the revised abstract is still a bit unclear. Please trim the length somewhat and add an introduction to the problem that you are trying to address. A revised abstract will make the contribution of this paper much clearer to readers.

---

## Round 0.4 · accepted · Accept

Thanks for these revisions - the abstract is much approved.